# Christmas Tree Bio-Waste as a Power Source of Bioactive Materials with Anti-Proliferative Activities for Oral Care

**DOI:** 10.3390/molecules27196553

**Published:** 2022-10-03

**Authors:** Bartosz Tylkowski, Piotr Konopka, Malgorzata Maj, Lukasz Kazmierski, Monika Skrobanska, Xavier Montane, Marta Giamberini, Anna Bajek, Renata Jastrzab

**Affiliations:** 1Eurecat, Centre Tecnològic de Catalunya, Marcelli Domingo s/n, 43007 Tarragona, Spain; 2Faculty of Chemistry, Adam Mickiewicz University in Poznan, Uniwersytetu Poznanskiego 8, 61-614 Poznan, Poland; 3Tissue Engineering Department, Chair of Urology and Andrology, The Ludwik Rydygier Collegium Medicum in Bydgoszcz, Nicolaus Copernicus University in Torun, Karlowicza str 24, 85-092 Bydgoszcz, Poland; 4Departament de Química Analítica i Química Orgànica, Facultat de Quimica, Universitat Rovira i Virgili, Carrer Marcelli Domingo s/n, 43007 Tarragona, Spain; 5Department of Chemical Engineering, University Rovira i Virgili, Av. Països Catalans 26, Campus Sescelades, 43007 Tarragona, Spain

**Keywords:** bio-waste, anticancer, oral diseases, flavonoids, luteolin

## Abstract

According to the American Cancer Society, roughly 54,000 new cases of oral cavity or oropharyngeal cancers have been detected in the United States of America in 2021, and they will cause about 10,850 deaths. The main therapies for cancer management, such as surgery and radio- and chemotherapy, have some own benefits, albeit they are often destructive for surrounding tissues; thus, deep investigations into non-surgical treatments for oral cavities are needed. Biologically active compounds (BACs) extracted from European Spruce needles were analyzed to determine the total phenolic and flavonoid content and were used as additional ingredients for oral hygiene products. An anti-proliferation investigation was carried out using extracts containing BACs with the use of several cell lines (cancer and a normal one). ESI-MS studies on BACs showed that luteolin, a natural flavonoid compound with anti-tumorigenic properties against various types of tumors, is the predominant component of the extracts. MTT, BrdU, and LIVE/DEAD studies demonstrated that BAC extracts obtained from Christmas tree needles possess anticancer properties against squamous cell carcinoma (with epithelial origins). We proved that BAC extracts contain high amounts of luteolin, which induces cytotoxicity toward cancer cells; along with their high selectivity, robustness, and nontoxicity, they are very promising materials in oral health applications.

## 1. Introduction

Cancer is as a leading cause of death and, globally, is a major barrier to increasing life expectancy. According to a recent press announcement by the American Cancer Society, in 2021, approximately 54,000 new cases of oral cavity or oropharyngeal cancers have been detected in the United States of America, and they will cause about 10,850 deaths [1]. Most people diagnosed with these sorts of cancers are, on average, 63 years old. Only over 20% (1 in 5) of cases occur in patients younger than 55 years old [2]. Published data also highlight that these cancers are more than twice as common in men as in women. Sung and co-workers examined the cancer burden worldwide in 2020 based on the GLOBOCAN estimates of cancer incidence and mortality produced by the International Agency for Research on Cancer. According to published data in 2020, worldwide, 177,757 people passed away from cavity cancer, while oropharyngeal cancer caused 48,143 deaths [2]. Consequently, the diagnoses of both cancers are challenging, especially as they have also been reported in nonsmokers and nondrinkers. The main therapies for cancer management, such as surgery and radio- and chemotherapy, have some benefits, albeit they are often destructive to surrounding tissues [3]. Therefore, non-surgical treatments for oral cavities are needed. Biologically active compounds (BACs), such as polyphenols and flavonoids [4], are well known in nonconventional medicine (i.e., herbal medicine), due to their anticancer, anti-inflammatory, antibacterial, and antioxidant properties [5]. Several papers describe the beneficial properties of polyphenols extracted from pine bark, such as the anti-viral (HIV-1) [6], neuroprotective [7], antioxidant, anti-inflammatory, and anti-diabetic properties [8]; however, there is a knowledge gap on BAC extracts obtained from *European Spruce* needles. 

Thus, the objective of our work was to carry out solid–liquid extractions of BCAs from European Spruce *Picea abies* needles and to investigate their anti-proliferative properties as a potential tool in oral care treatment. During the solid–liquid extraction, we used phosphate-buffered saline (PBS) as a solvent to implement the ISO 10993-5:2009 regulation to access in vitro cytotoxicity tests. Furthermore, considering that more than 90% of oral cancers are squamous cell carcinoma (SCC) with epithelial origin [3], our studies were performed using isolated epidermoid carcinoma cells. Fallen bio-waste needles of Christmas tree, as the sources of anti-proliferative BACs, were selected. The dropped needles were collected from a European Spruce *Picea abies* tree, which is one of the most popular Christmas trees in East–Central Europe, due to its low market price [9]. Each year, about 33 to 36 million Christmas trees are produced in the USA and 50 to 60 million trees are produced in Europe before the Christmas period; most are thrown in trash cans as waste. Therefore, they could represent valuable (but extremely cheap) sources of BACs.

## 2. Results and Discussion

In vitro cell viability and cytotoxicity assays with cultured cells have been widely applied in oncological research to evaluate both compound toxicity and tumor cell growth inhibition during drug administration. To fulfill the ISO 10993-5:2009 regulations, a BAC solid–liquid extraction with PBS was performed. As described in the methodology section, conventional extraction experiments were carried out for up to 24 h. High equilibrium values of the extracted species were achieved, i.e., 24 mg/(g solid) for total phenolics and 21 mg/(g solid) for total flavonoids, respectively. 

Figure 1 shows the extraction kinetics with PBS as the solvent, which was followed by measuring the liquid phase concentrations of polyphenols and flavonoids for 24 h. It seems that the maximum TPC was reached after 5 min of extraction with PBS, while 90% of TFC maintained constant (up to 1 h of extraction), with a slight increase during the next 23 h to reach a constant value. To verify the influence of the solvent type on TPC and TFC, an additional 1 h of the extraction experiment was performed with water and methanol as solvents. The corresponding results are shown in Figure 2. 

The influence of PBS on the BAC extraction efficiency from apiculture products was recently investigated [10]. During a solid–liquid extraction process of BACs from propolis, the authors replaced water with PBS at pH 6.75. Their results demonstrated that the concentration of polyphenols was higher in the extract obtained with PBS than in the aqueous one. In order to verify the impact of PBS on BAC extraction from *Picea abies*-grounded needles, solid–liquid extractions with water at pH 4.75 were carried out at the same pH (as in the case of PBS). This value was indicated in the ISO 10993-5:2009 (2021) regulation as the optimum one for investigation with cell lines [11]. The results showed that total polyphenol and flavonoid concentrations in the PBS extracts were slightly higher (approximately 5%) than in the aqueous extracts. In order to check the effect of pH on BAC content, a supplementary experiment was performed with PBS at pH 7.00. A comparison of the results obtained at pHs 4.75 and 7.00 showed that BAC extraction at pH 4.75 was slightly better. To verify the effect of the grinder on the BAC content in the extract, 1 and 24 h extractions with entire needles were also performed at room temperature with the PBS at pH 4.75. 

As shown in Figure 3, after 24 h of BAC extraction from entire needles, only 0.95 mg/g and 0.32 mg/g of TPC and TFC were recovered. This means that extraction efficiency values decreased 25 and 67 times for TPC and TFC, respectively. It is well known that the use of alcoholic solvents leads to maximum BAC extraction plant materials, apiculture products, and bio-waste. Indeed, by using methanol as the solvent, 47 mg of TPC and 37 mg of TFC were extracted from 1 g of grounded material. Thus, the extraction efficiency was 2-fold higher for TPC and 1.5-fold higher for TFC in comparison to the PBS extracts (see Figure 2) investigating BAC extractions from frozen needles of the Norway spruce (*Picea abies*) via boiling methanol [12]. By applying the Folin–Ciocalteu method, the authors reported that the average concentrations of total phenolics in the needles of different ages and damage classes were between 58 and 81 mg in 1 g of needles frozen in liquid nitrogen. Moreover, the authors reported that no significant differences were found in the concentrations of phenolics in needles from trees of different damage classifications. It is important to highlight that by PBS selection, as with the extraction solvent, on the one hand, we ensured that the solvent itself did not affect the results of cell viability tests and, on the other hand, the obtained extract could be directly applied as a part of an innovative alcohol-free mouthwash formulation for oral health care treatment. ESI-MS studies were performed to identify the BAC compositions in aqueous and methanol extracts.

Table 1 presents the results of the qualitative analysis of compounds of the extracts [13,14,15]. The ESI-MS analysis shows that the obtained extracts are mixtures of different types of phenolic compounds. Based on the identification of fragment ions and their corresponding molecular weights, four distinct phenolic compounds were identified. As can be seen in Figure 4, the predominant one is luteolin, a natural flavonoid compound, which has been shown to have anti-tumorigenic properties against various types of tumors, including the most common malignancy of the oral cavity: oral squamous cell carcinoma [16]. Considering that more than 90% of oral cancers are squamous cell carcinoma (SCC) with epithelial origins, the A-431 human cell line was used in the following experiment, which was established from epidermoid carcinoma with epithelial morphology [3]. Moreover, it can serve as a model in oral cancer research. To investigate the cytotoxic activity of the tested compounds, the metabolic activities of both cell lines were measured (Figure 5A).

After 24 h of incubation, only minor changes in mitochondrial dehydrogenase activity were found; 72 h of incubation with 20-fold diluted BAC-PBS extracts reduced ASC viability by 17.7% (*p* < 0.01) compared to the control. Interestingly, cancer cell viability reduced by 40.7% (*p* < 0.0001) when they were analyzed under the same conditions. As aforementioned, six different natural compounds can be detected in the BAC-PBS extract, in which anti-proliferative and anticancer potential are documented in the available literature. However, luteolin showed the highest concentration; therefore, our study focused on this natural agent in the following discussion. Tjioe et al. (2016) showed that luteolin reduces the viability of the human tongue squamous carcinoma-derived cell line (SCC-25) in the concentration of 10 µM. It was noticed that the number of actively proliferating cancer cells reduced by 25% during the first 24 h. As measured using the thymidine analog, the total cell number decreased to 53%. No changes were observed for ASCs [16]. Similar to the studies, this research also demonstrates a minor toxicity in normal cells. To our knowledge, the investigation on luteolin toxicity to oral cancer has not been deeply carried out, although other studies have documented that this compound decreases cell viability in different cancer cells. Luteolin in concentrations of 60 and 100 µM reduces the viability of cancer gastric cells (CRL-1739) by more than 50% [17,18]. Furthermore, regarding the same effect on lung cancer cell growth, Raina et al. (2021) proved that such treatment exhibited cytotoxicity towards HeLa cells in dose- and time-dependent manners [17,19]. Many studies have documented that luteolin generally decreased cell numbers over time, which strongly suggests a cytotoxic effect. Luteolin can reduce the migratory features of oral cancer cells, which is very promising in the case of the BAC-PBS extracts evaluated by our group [16,20,21].

The objective of our research was to distinguish living cells from dead cells using a cell-permeable dye for staining live cells and a cell-impermeable dye for staining dead and dying cells, which were characterized by damaged cell membranes. Live cells exhibit intracellular esterase activity as determined by the enzymatic conversion of non-fluorescent calcein-AM to the intensely fluorescent calcein, which is well-retained within live cells. It was observed that BAC-PBS extract reduced ASC and A-431 cell viability after 72 h by, respectively, 38.2% (*p* < 0.01) and 53.4% (*p* < 0.001) in comparison to the control (Figure 4B). Representative images of stained cells are shown in Figure 4C.

To detect the newly synthesized DNA of actively proliferating cells, thymidine analogue and Alexa Fluor 488-conjugated anti-BrdU monoclonal antibody were used. Co-staining with DAPI allowed for quantification of the total cell numbers. It was observed that 72 h incubation with BAC-PBS extract reduced proliferation of A-431 cells by 26.7% in comparison to the control (Figure 4D). Furthermore, cell numbers were reduced by 46.8%, which could be attributed to the luteolin effect from BAC-PBS. Recently, Ho et al. (2021) noticed that this compound significantly reduced the proliferation of the nasopharyngeal cancer cell line (NPC), evidence that was also demonstrated by other researchers [22,23,24]. Interestingly, no effects were observed for ASCs. In this case, both proliferation and cell numbers were comparable to the control. An analogous effect was noticed after the luteolin treatment of lung cells (cancer and normal ones) [22]. They demonstrated that luteolin (in concentrations of 20 or 40 µmol/L) inhibited the proliferation of cancer cells (but not the normal ones) in a time-dependent manner.

## 3. Materials and Methods

The needle biomass was from a European Spruce used as Christmas tree during the 2021 Christmas period. Samples of the biomass were collected and dried in an oven at 30 °C until constant weight was observed. Then, needles were ground using a knife mill Retsch GRINDOMIX GM 200 (Hann, Germany), obtaining a final fineness < 300 µm.

### 3.1. Biologically Active Compound Extractions

Freshly-ground needles and entire needles were used for biologically active compound (BAC) extractions. Solid–liquid extractions were performed using 1 g of biomass material with 20 mL of phosphate-buffered saline (PBS) at pH 4.75 for 5 min, 15 min, 30 min, 1 h, 2 h, 6 h, and 24 h at room temperature (22 ± 2 °C). PBS was selected as the solvent to implement the ISO 10993-5:2009 regulation to access in vitro cytotoxicity tests. For sake of comparison, additional extraction experiments with water, water at pH 4.75, and methanol were carried out for 1 h. All extractions with ground samples were carried out by magnetic stirring at 750 rpm, while the extraction of uncrushed needles was performed using the Orbital Shaker Incubator ES 80 Noki (Zhengzhou, China) set up at 250 rpm, due low wettability. After performing the extractions, the mixtures were filtered using sterile syringe filters containing a polyether sulfone membrane with a 0.22 µm pore size (VWR). Finally, the extracts were analyzed to determine the total phenolic and flavonoids content, polyphenol compositions, and anticancer properties.

### 3.2. Total Phenolic Content

The total phenolic content (TPC) was determined by using the Folin–Ciocalteu method as described in the literature [23,24]. The following protocol was applied: 250 µL of diluted extract was added to the cuvette containing 250 µL of the Folin–Ciocalteu reagent and 1.25 µL of ultrapure water. After 5 min, 375 µL of 20% Na_2_CO_3_ and 1.00 µL of ultrapure water were added. The absorbance was measured at 765 nm after 30 min using a UV-Vis Spectrometer. PBS at pH 4.75 was used as a blank solution. TPC is reported as the gallic acid (3,4,5-trihydroxybenzoic acid) equivalent calculated by the following equation: Abs = 0.00861, C + 0.0441, R^2^ = 0.998. Sodium carbonate, gallic acid, as well as the Folin–Ciocalteu phenolic reagent were supplied by Merck. PBS (free of calcium and magnesium ions) was provided by Corning. The pH of the PBS to 4.75 was adjusted by using hydrochloric acid supplied by Merck.

### 3.3. Total Flavonoid Content

The total flavonoid content (TFC) was determined by using the method involving the aluminum chloride complex formation [25]. In this procedure, 0.5 mL of 5% AlCl_3_ was added to the 25.0 mL volumetric flask containing 10.0 mL of methanol. Then, 1.0 mL of the test solution was added to the flask and the volume was made up with methanol to 25.0 mL. The absorbance was measured at 300 nm after 30 min by means of UV-Vis Spectrometer. PBS at pH 4.75 was used as the blank solution. TFC is reported as the quercetin (3,3′,4′,5,7-pentahydroxyflavone) equivalent, calculated by the following equation: Abs = 2.5022 × C; R^2^ = 0.9875. Aluminum chloride anhydrous, quercetin, and methanol were supplied by Merck.

### 3.4. Electrospray Mass Spectra Analysis

Electrospray mass spectra (ESI-MS) were performed in methanol using a Waters Micromass ZQ mass spectrometer (Agilent Technologies, Santa Clara, CA, USA). The samples were run in a positive-ion mode. Scanning parameters were from *m/z* = 100 to 1000 in 6 s, and 10 scans were added up to obtain the spectrum.

### 3.5. Cell Culture

Anticancer investigations were carried out using PBS extracts containing BACs (BACs-PBS). The following cell lines and cells cultures were used in the studies.

Epidermoid carcinoma cells (A-431, cat. no. CRL1555) and mesenchymal stem cells immortalized with hTERT (ASC52telo, cat. no. SCRC-4000) were obtained from American Type Culture Collection (ATCC). Carcinoma cells, derived from the epidermis of an 85-year-old female, showed epithelial morphology. They are highly tumorigenic and were transplanted into immunosuppressed mice to form rapidly growing subcutaneous tumors. ASC52telo cells were obtained from adipose tissue of a female donor. They showed fibroblast-like morphology and high expressions of characteristic surface markers: i.e., CD29, CD44, CD73, CD90, CD105, and CD166 (>90%), and low expressions of CD14, CD19, CD34, and CD45 (<5%). Cells maintained multipotent phenotypes with the ability to differentiate into adipocytes, osteoblasts, and chondrocytes (as verified at ATCC).

All basic cell culture reagents were purchased from Corning. Cells were cultured in Dulbecco’s Modified Eagle Medium (DMEM) supplemented with 10% FBS. Additionally, medium for ASC was supplemented with 10 ng/mL bFGF (Thermo Fisher Scientific, Waltham, MA, USA). Cells were maintained in a humidified incubator (Thermo Electron Corporation) with 5% CO_2_ at 37 °C. The standard growth medium was changed every 2–3 days.

### 3.6. MTT Assay

Thiazolyl blue tetrazolium bromide (MTT, Sigma-Aldrich, Saint Louis, MO, USA) was used to measure cell viability as a function of the redox potential according to ISO 10993-5:2009. Actively respiring cells convert the water-soluble MTT to an insoluble formazan. Cells were seeded on 96-well plates at a density of 5000 cells/cm^2^. After preincubation during 24 h, compounds were added to the selected wells, and cells were cultured for an additional 24–72 h. The medium was then discarded, and cells were incubated for 1 h at 37 °C with 1 mg/mL MTT solution in DMEM without phenol red. After incubation, formazan crystals were dissolved in DMSO (StanLab). The absorbance was read at 590 nm on a microplate reader (Multiskan Sky, Thermo Fisher Scientific, Waltham, MA, USA).

### 3.7. LIVE/DEAD Assay

LIVE/DEAD™ Cell Imaging Kit (Thermo Fisher Scientific, Waltham, Massachusetts, USA) was used to determine cell viability based on esterase activity and plasma membrane integrity. Simultaneously staining with green-fluorescent calcein-AM (ex/em 488 nm/515 nm) and red-fluorescent DNA stain (ex/em 570 nm/602 nm) allows for precise determination of live and dead cells in a population. Cells were seeded on 96-well plates at a density of 5000 cells/cm^2^. After preincubation for 24 h, compounds were added to selected wells, and cells were cultured for an additional 72 h. According to the manufacturer’s protocol, an equal volume of a 2× working solution (mixture of comp. A and B) was then pipetted, and after 20 min of incubation at RT, wells were rinsed with PBS. Labeled cells were viewed under a fluorescence microscope (IX83, Olympus, Tokyo, Japan). Microscopic images were used for subsequent quantitative analysis (cell Sens Dimension, Olympus, Tokyo, Japan).

### 3.8. BrdU Assay

Cell proliferation was measured using thymidine analogue 5-Bromo-2’-deoxyuridine (BrdU, Sigma-Aldrich, Saint Louis, MO, USA) following its incorporation into newly synthesized DNA and subsequent detection with an anti-BrdU antibody. Cells were seeded on 96-well plates at a density of 5000 cells/cm^2^. After a 24 h period of pre-incubation, compounds were added to selected wells, and cells were cultured for additional 72 h. BrdU was added 24 h before the end of the experiment. Cells were fixed at RT using 4% methanol-free formaldehyde with 0.1% Triton X-100 in PBS for 15 min and washed with fresh PBS afterwards. Then, they were incubated with 2 M HCl, and finally with 0.01 M phosphate/citric acid buffer. Following acid-washing, 4% BSA buffer with 0.1% Triton X-100 in PBS was used for blocking with an incubation carried out overnight. Cells were stained at RT with Alexa Fluor 488 conjugated anti-BrdU monoclonal antibody (Thermo Fisher Scientific, Waltham, MA, USA) for 1 h and counterstained with 1 μg/mL DAPI for 15 min. Labeled cells were imaged using an inverted, motorized, fluorescence microscope (IX83, Olympus). Both the source light power and acquisition times were calibrated using the appropriate controls. The proliferation rate was calculated using a % of total cells (DAPI positive) compared to divided cells (BrdU positive). Image data were analyzed using HCS ScanR software provided for a demo period by Olympus Poland. Object detection was carried out using an edge detection algorithm and the segmentation data from DAPI channel were used as the bases for calculating the total intensities of Alexa Fluor 488 for those objects.

## 4. Conclusions

To summarize, we successfully proved that BAC-PBS extract, which contains a high amount of luteolin, induces cytotoxicity toward cancer cells, which, together with its high selectivity, robustness, and nontoxicity, make them very promising materials for oral health applications. In addition, in our previous studies, we fine-tuned an extraction procedure that produced a BAC extract, which can be directly used for in vitro tests and in the formulation of alcohol-free mouthwashes. It is a step forward in clinical application, in which a combination of different methods can be applied in oral cancer treatment. Our future research will also focus on the applicability of BAC-PBS extracts as adjuvant anticancer therapies.

## Figures and Tables

**Figure 1 molecules-27-06553-f001:**
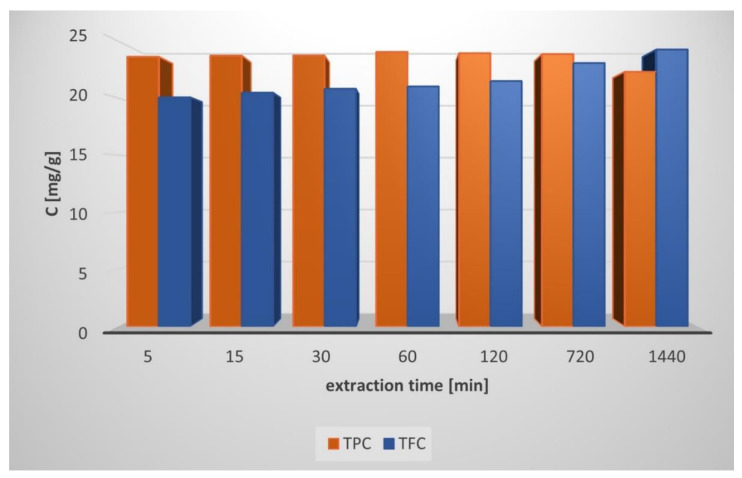
Total phenolic content (TPC) and total flavonoid content (TFC) vs. the time of extraction.

**Figure 2 molecules-27-06553-f002:**
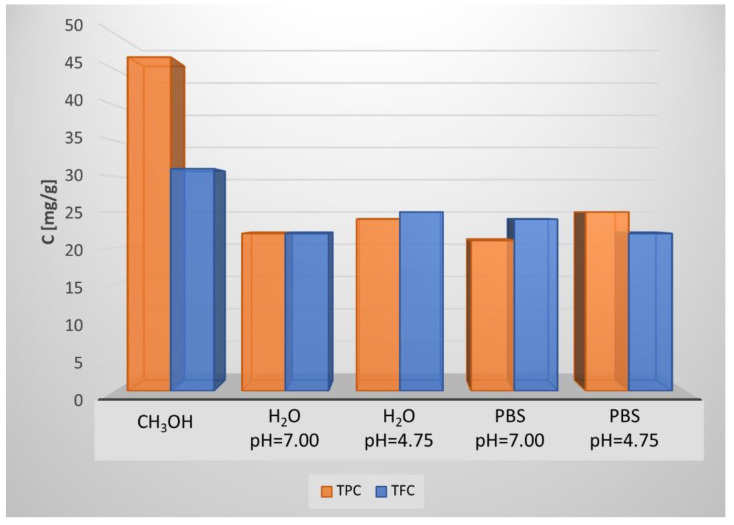
Total phenolic content (TPC) and total flavonoid content (TFC) vs. the type of solvent (a 1 h time of extraction).

**Figure 3 molecules-27-06553-f003:**
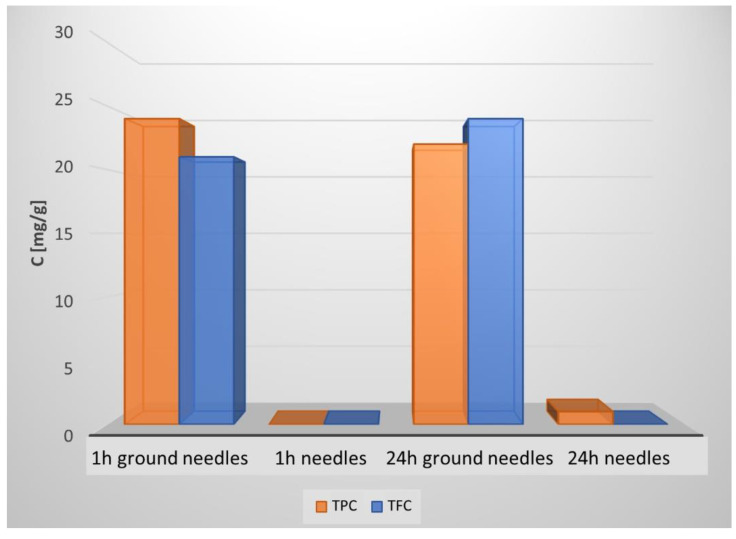
Comparisons of total phenolic content (TPC) and total flavonoid content (TFC) extracted from ground needles and non-ground needles (1 and 24 h extraction times).

**Figure 4 molecules-27-06553-f004:**
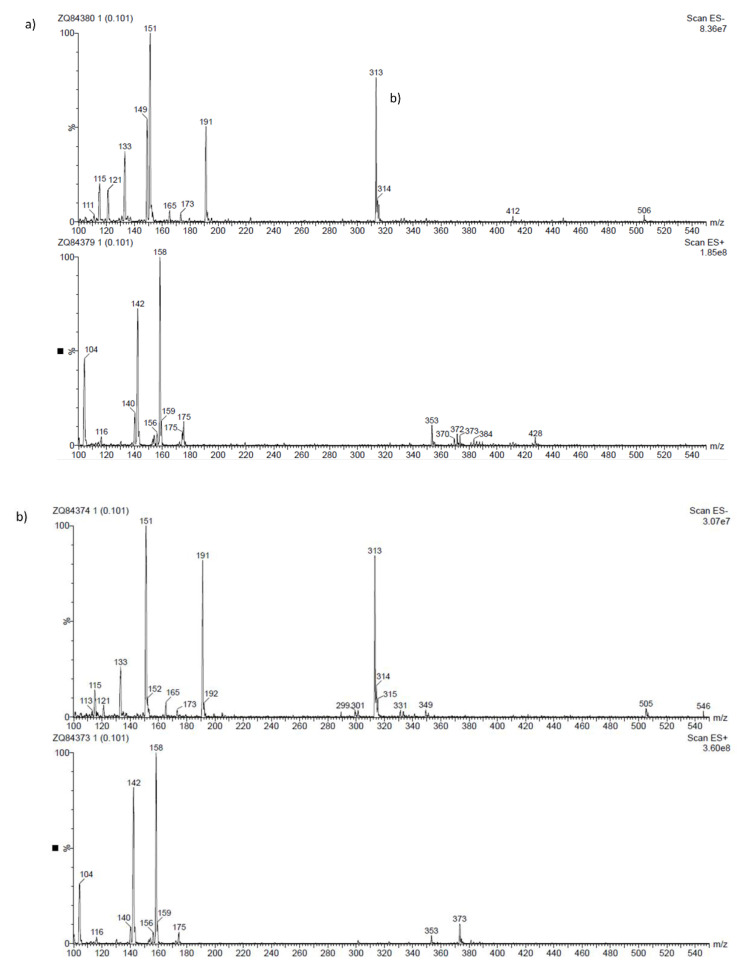
ESI-MS spectra of extract from European Spruce, (**a**) water extract, (**b**) methanol.

**Figure 5 molecules-27-06553-f005:**
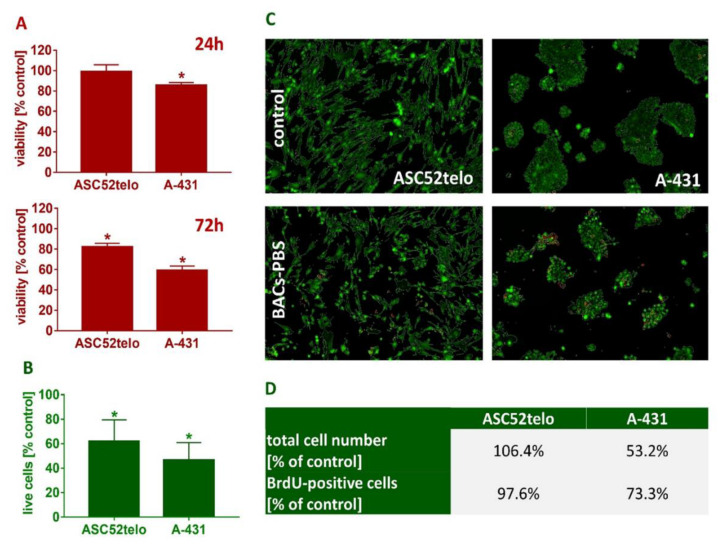
BACs extract influences on proliferation and viability of cultured cells. (**A**) ASCs and A-431 viability after incubation with BACs extract. (**B**) Esterase activity in cells cultured with the extract. Bars represent standard deviation; * Denotes significant difference at *p* < 0.05. (**C**) Representative images of ASCs and A-431 cells cultured for 72 h in 20-fold diluted BACs extract. As analyzed by dedicated software, green stands for live cells and red for dead cells. (**D**) Proliferation of ASCs and A-431 cells cultured with the extract.

**Table 1 molecules-27-06553-t001:** The qualitative analysis of BAC compounds in the extracts obtained by means of ESI-MS.

Compound	Chemical Formula	Molecular Mass g/mol	Identification Fragment Ions	Structure
Luteolin	C_15_H_10_O_6_	286.24	*m/z* 151*m/z* 133*m/z* 121*m/z* 107	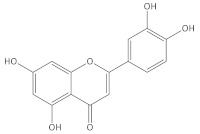
Pilloin	C_17_H_14_O_6_	314.29	*m/z* [M+K]^+^ 353*m/z* [M−H]^−^ 313	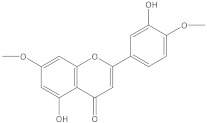
*o*-Vanillin	C_8_H_8_O_3_	152.15	*m/z* [M−H]^−^ 151	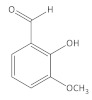
Quinic acid	C_7_H_12_O_6_	192.17l	*m/z* [M−H]^−^ 191	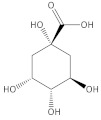
Benzoic acid	C_7_H_6_O_2_	122.12	*m/z* [M−H]^−^ 121	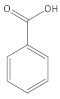
Isocitric acid	C_6_H_8_O_7_	192.12	*m/z* [M−H]^−^ 191*m/z* 173*m/z* 155*m/z* 111	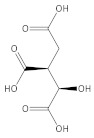

## Data Availability

Not applicable.

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
