# Peer review of "Christmas Tree Bio-Waste as a Power Source of Bioactive Materials with Anti-Proliferative Activities for Oral Care"

_molecules, 2022, doi:10.3390/molecules27196553_

Round 1

Reviewer 1 Report

Authors of manuscript described the content of Christmas-tree needles and the anticancer potential of this extract against cancer cell line A-431 and fibroblast-like cell line ASC52telo. It is interesting work but I have a few comments:

Line 19, 22 and 26: It should be without semicolon at the end of sentence.

Figure 2: Authors present extraction by different solvents after 1 hour, and on the same figure we can see results of needles extraction using PBS after 24 hours (what was the pH ?). I understand that Authors want to present the comparison of extraction using PBS after 1 and 24h but it should be on another graph/table/figure, because in this way it is false title under the figure, that all extractions was after 1hour. Furthermore, is the concentration of TPC and TFC correct on figure 2? Because on figure 1 the content of TPC and TFC after 12 and 72 hour is about 20-25 mg/g and after 24h below 5 mg/g (figure 2)? Please explain it.  Moreover, the title below the figure should be changed, because eluent is using in chromatography but in extraction solvents are using.

Table 1: Please improve the structure of pillion, it is 3’,5-dihydroxy-4’,7-dimethoxyflavone, the Authors of manuscript drawn 4’,7-dihydroxy-3’,5-dimethoxyflavone. Furthermore, third compound should be named as o-vanillin.

Line 125-126: The sentence: “Based on the identification of fragment ions and their corresponding molecular weights, 6 distinct phenolic compounds were identified.”. According to the table 1, in extract, there are 4 phenolic compounds (with aromatic ring). Please improve it.

Line 146: Please improve the concentration unit. “the concentration of 10 …?”, it should be 10 µM? The same situation is in line 152. Please improve it.

Line 199-202: Please verify the units of all substances. What was the final volume of cuvette ? Because, when Authors used 250mL of extract and 250mL of Folin-Ciocalteu reagent it is 0,5 liter in cuvette. In my opinion it should be 250 microliter (µL) and then 375 µL of Na2CO3? Please explain it.

Line 210: "AlCl3", “3” should be in subscript, it should be “AlCl3”.

In section Materials and Methods it would be more readable to use subsections with different methods, e.g. extraction, TPC content, TFC content, culture medium, MTT assay, LIVE/DEAD assay etc.

Author Response

#1

The first of all we would like to thank you for all remarks which improved our paper.

Authors of manuscript described the content of Christmas-tree needles and the anticancer potential of this extract against cancer cell line A-431 and fibroblast-like cell line ASC52telo. It is interesting work but I have a few comments:

Line 19, 22 and 26: It should be without semicolon at the end of sentence.

Semicolons were deleted.

Figure 2: Authors present extraction by different solvents after 1 hour, and on the same figure we can see results of needles extraction using PBS after 24 hours (what was the pH ?). I understand that Authors want to present the comparison of extraction using PBS after 1 and 24h but it should be on another graph/table/figure, because in this way it is false title under the figure, that all extractions was after 1hour. Furthermore, is the concentration of TPC and TFC correct on figure 2? Because on figure 1 the content of TPC and TFC after 12 and 72 hour is about 20-25 mg/g and after 24h below 5 mg/g (figure 2)? Please explain it.  Moreover, the title below the figure should be changed, because eluent is using in chromatography but in extraction solvents are using.

Thank you for this comment. We additionally wanted to show the differences between extraction from ground and non-ground needles. Unfortunately, as the reviewer mentioned, it was unclear in the graph. Figure 2 has been revised and we have additionally created a new graph (Figure 3) with a comparison of extraction from ground and non-ground needles. Title of Figure 2 was changed, sorry for our mistake.

Table 1: Please improve the structure of pillion, it is 3’,5-dihydroxy-4’,7-dimethoxyflavone, the Authors of manuscript drawn 4’,7-dihydroxy-3’,5-dimethoxyflavone. Furthermore, third compound should be named as o-vanillin.

The structured of pillion was changed and name o-vanillin was corrected.

Line 125-126: The sentence: “Based on the identification of fragment ions and their corresponding molecular weights, 6 distinct phenolic compounds were identified.”. According to the table 1, in extract, there are 4 phenolic compounds (with aromatic ring). Please improve it.

It was corrected.

Line 146: Please improve the concentration unit. “the concentration of 10 …?”, it should be 10 µM? The same situation is in line 152. Please improve it.

The concentration unit was improved.

Line 199-202: Please verify the units of all substances. What was the final volume of cuvette ? Because, when Authors used 250mL of extract and 250mL of Folin-Ciocalteu reagent it is 0,5 liter in cuvette. In my opinion it should be 250 microliter (µL) and then 375 µL of Na2CO3? Please explain it.

Mistake of the units was corrected.

Line 210: "AlCl3", “3” should be in subscript, it should be “AlCl3”.

It was corrected.

In section Materials and Methods it would be more readable to use subsections with different methods, e.g. extraction, TPC content, TFC content, culture medium, MTT assay, LIVE/DEAD assay etc.

Titles of the subsection were added in the Section Materials and Methods for improve readable of the text.

Reviewer 2 Report

This is an interesting study whose topic coincides with the scope of the journal.
This study focuses on the investigation of the anti-proliferative properties of biologically active

compound from Christmas trees as a potential tool in oral care treatment.

The obtained results showed that biologically active compound extract, mainly Luteolin,
induces cytotoxicity toward cancer cells.

I found this paper rich both in the experimental part and the theoretical part. However, there
are some remarks about this paper:

Point 1: Concerning the extraction of biologically active compound, authors have to mention
the reasons of their choice of the extraction method and parameters (solvent, pH ...). I haven’t
seen also any reference for the extraction method in the experimental part.

Point 2: Authors indicated in the results section that they carried out an ESI-MS analysis, but
this analysis was not showed in the experiment part.

In addition, the ESI-MS spectra (Figure 3) is not clear. Authors should improve the quality of
the figure.

Point 3: Authors cited that the predominant compound was Luteolin. I encourage authors to
perform a quantitative analysis to ensure this result.

Author Response

#2

This is an interesting study whose topic coincides with the scope of the journal.
This study focuses on the investigation of the anti-proliferative properties of biologically active compound from Christmas trees as a potential tool in oral care treatment. The obtained results showed that biologically active compound extract, mainly Luteolin, induces cytotoxicity toward cancer cells.
I found this paper rich both in the experimental part and the theoretical part. However, there are some remarks about this paper:
Point 1: Concerning the extraction of biologically active compound, authors have to mention the reasons of their choice of the extraction method and parameters (solvent, pH ...). I haven’t seen also any reference for the extraction method in the experimental part.

Information on the choice of procedure has been added to the experimental section.

Freshly ground needles and entire needles were used for BACs extraction. Solid-liquid extractions were performed using 1g of biomass material with 20 mL of Phosphate-Buffered Saline (PBS) at pH 4.75 for 5 min, 15 min, 30 min, 1 h, 2 h, 6 h and 24 h at room temperature (22 ± 2 °C). PBS was selected as a solvent to implement the ISO 10993-5:2009 regulation to access in vitro cytotoxicity tests. For sake of comparison, additional extraction experiments with water, water at pH 4.75 and methanol were carried out for 1h.

Point 2: Authors indicated in the results section that they carried out an ESI-MS analysis, but
this analysis was not showed in the experiment part.

Information about ESI-MS analysis was added to the Experimental part.

In addition, the ESI-MS spectra (Figure 3) is not clear. Authors should improve the quality of the figure.

The quality of the ESI-MS spectra were improved.

Point 3: Authors cited that the predominant compound was Luteolin. I encourage authors to perform a quantitative analysis to ensure this result.

This is a very valuable remark. Comparison of the activity of pure luteolin with the obtained Christmas tree extract will certainly help to provide information on whether there is a synergistic effect on the antiproliferation of Luteolin due to the co-presence of other phenols and flavonoids in the Christmas tree extract. Unfortunately, due to the short response time for reviews, we are unable to conduct a comparison of the antiproliferative activity of the BACs extract with Luteolin alone in this work. This will be included in the next paper, which is in preparation and will deal with encapsulation of Christmas tree extract in chitosan.

Reviewer 3 Report

My comments are below, expanding from general methodological concerns to more targeted thoughts about the manuscript.1- It is unclear which is the gap of knowledge that this study aims to fill. What is the main important outcome of this research that we didn't know before? This should be clearly stated in the Abstract and the last paragraph of the Introduction.

2. More  experimental data are needed for BACs extract  to evaluate the antiproliferative activity of BACs extract such as dose responsive curves and IC50 and etc.   

 3. The authors have to compare the antiproliferative activity of  BACs extract with Luteolin alone in order to see if there is a synergistic effect on antiproliferative of Luteolin due to the copresence of other  phenolic and flavonoids in the christmas tree extract.

4.  Unit of concentration in page 6, line 146  is not clear. Is it molar or micromolar?  

  5.  Text has a grammatical and syntax errors as well as typos.

In short, the manuscript needs to be significantly revised from methodological perspective, presentation of results and the overall quality of the textbefore to be consider for publications in molecules.

Author Response

#3

My comments are below, expanding from general methodological concerns to more targeted thoughts about the manuscript.

1. It is unclear which is the gap of knowledge that this study aims to fill. What is the main important outcome of this research that we didn't know before? This should be clearly stated in the Abstract and the last paragraph of the Introduction.

They are several published papers on polyphenols extraction from pine bark extracts and their beneficial properties, such as: Anti-viral (HIV-1) Feng, W.Y.; Tanaka, R.; Inagaki, Y.; Saitoh, Y.; Chang, M.O.; Amet, T.; Yamamoto, N.; Yamaoka, S.; Yoshinaka, Y. Pycnogenol, a Procyanidin-Rich Extract from French Maritime Pine, Inhibits Intracellular Replication of HIV-1 as Well as Its Binding to Host Cells. Jpn. J. Infect. Dis. 2008, 61, 279–285.; Neuroprotective Sharma, A.; Goyal, R. In-Vitro Neuronal Cell Proliferation and in-Vivo Neuroprotective Activity of Pinus roxburghii for Memory and Cognition. Alzheimer’s Dement. 2020, 16, e038337., Antioxidant, anti-inflammatory, and anti-diabetic properties. Shand, B.; Strey, C.; Scott, R.; Morrison, Z.; Gieseg, S. Pilot Study on the Clinical Effects of Dietary Supplementation with Enzogenol, a Flavonoid Extract of Pine Bark and Vitamin C. Phytother. Res. 2003, 17, 490–494. However, there is a gap of knowledge of BACs extracts obtained from European Spruce needles. Thus, the aim of our work is to investigate the anti-proliferative properties of BACs from European Spruce as a potential tool in oral care treatment.

2. More  experimental data are needed for BACs extract  to evaluate the antiproliferative activity of BACs extract such as dose responsive curves and IC50and etc.   

The extensive biological analyses of the BACs extracts will be performed in the next steps of the experiment and are the aim of the further papers. It is true that information about IC10, 50 and 90 and also about e.g. apoptosis or other biological aspects are important and we are aware that needed in the future in the aspects of the potential clinical use. 

3. The authors have to compare the antiproliferative activity of  BACs extract with Luteolin alone in order to see if there is a synergistic effect on antiproliferative of Luteolin due to the copresence of other  phenolic and flavonoids in the christmas tree extract.

This is a very valuable remark. Comparison of the activity of pure luteolin with the obtained Christmas tree extract will certainly help to provide information on whether there is a synergistic effect on the antiproliferation of Luteolin due to the co-presence of other phenols and flavonoids in the Christmas tree extract. Unfortunately, due to the short response time for reviews, we are unable to conduct a comparison of the antiproliferative activity of the BACs extract with Luteolin alone in this work. This will be included in the next paper, which is in preparation and will deal with encapsulation of Christmas tree extract in chitosan.

4. Unit of concentration in page 6, line 146  is not clear. Is it molar or micromolar?  

It was very interesting observation made from the presented paper and currently such studies are designed and carried out by our team. The first, obtained, results are very promising (also in combination with currently used chemotherapy) and will certainly be the aim of separate paper. 

5. Text has a grammatical and syntax errors as well as typos.

The text of the manuscript has been revised in accordance with these recommendations.

In short, the manuscript needs to be significantly revised from methodological perspective, presentation of results and the overall quality of the text before to be consider for publications in molecules.

The manuscript has been revised by a native speaker.

Round 2

Reviewer 3 Report

Comments to Authors: All of the concerns are addressed in the revised version and I can see significant improvement in the manuscript and I have no further comments. In my opinion the revised manuscript is now suitable for publication in its current form in Molecules.